# How Does the Geography Curriculum Contribute to Education for Sustainable Development? Lessons from China and the USA

**Sheng Miao [1], Michael E Meadows [2,3,4]****, Yushan Duan [1,\*] and Fengtao Guo [1,\*]**

1   School of Geographic Sciences, East China Normal University, Shanghai 200241, China
2   Department of Environmental Geographical Science, University of Cape Town, Rondebosch, Cape Town 7701, South Africa
3   College of Geography and Environmental Sciences, Zhejiang Normal University, Jinhua 321004, China
4   School of Geography and Ocean Science, Nanjing University, Nanjing 210023, China
\*   Correspondence: ysduan@geo.ecnu.edu.cn (Y.D.); ftguo@geo.ecnu.edu.cn (F.G.)

**Abstract:** Education for Sustainable Development (ESD) must play an important part in achieving the Sustainable Development Goals and, while it may be advanced through harnessing the unique advantages of the geography curriculum, connections between the geography curriculum and sustainable development competencies have not yet been systematically investigated in China and America. In order to explore the value of geography education in promoting learner competencies in sustainable development, we conducted a detailed analysis of China's geography curriculum standards and American geography curriculum standards, and explored the potential contribution of the geography curriculum to ESD. Learning objectives in China's geography curriculum standards for middle school (98 items) and high school (141 items), and American geography curriculum standards for middle school (80 items) and high school (85 items) were analyzed using content analysis supported by WordStat 8.0. Our findings suggest that geography education plays an important role in cultivating learners' cognition and ability regarding sustainable development, although there remains much room for improvement in cultivating learner attitudes and values towards ESD.

**Keywords:** ESD; geography curriculum standards; content analysis; learner competencies; middle school; high school

## 1. Introduction

The increasing scale and complexity of global environmental and sustainability challenges suggest that there is an urgent need for changes in formal education. A sustainable future cannot be realized unless there are developments in education across every sector so that all people have a clear awareness and understanding that the Earth's environment is the very basis for life. Indeed, the notion of environmental literacy is founded on the principle of providing broad, continuing, and repetitive exposure to environmental issues throughout the curriculum [1]. As one of the basic United Nations (UN) Sustainable Development Goals (SDGs Goal 4, "Quality Education", see: https://sdgs.un.org/goals, accessed on 25 September 2015), education today must be considered as an important factor in achieving sustainable development and, as a matter of fact, this idea was proposed as early as 2005 when the UN declared the Decade of Education for Sustainable Development (DESD) [2].

Sustainable development (SD), defined as "development that meets the needs of the present without compromising the ability of future generations to meet their own needs" [3] is now a global imperative. More than any other target, SDG 4.7 explicitly links ESD to other SDGs by stating that "all learners acquire the knowledge and skills needed to promote sustainable development". In order to cope with the global economic recession, climate change, and other sustainability challenges, China actively promotes the implementation

of the United Nations' 2030 Agenda for Sustainable Development with the goal of building a community with a shared future for humankind. This is reflected in a series of governmental documents, including China's Fourteenth Five-Year Plan, which details a vast array of the country's economic, social, and environmental priorities as well as Education Modernization 2035, which lays out the framework, including methods, content, and recommendations for the realization of SDG [4], aiming to ensure that learners understand what is meant by sustainable development, live a green and sustainable lifestyle, and promote a culture of peace, non-violence, global citizenship, and appreciation of cultural diversity [5]. Despite these clear signs of progress on both global and national scales, the impact of education towards a sustainable future remains constrained by many barriers, such as lack of adequate sources, insufficient support for in-service teachers, etc., and is all too often seen as an "add-on" element of the curriculum [6]. In this regard, novel forms of learning are entering the arena of ESD and environmental education more broadly, such as "transformative social learning", which requires the integrative switching back and forth among a set of mindsets [7]. Innovation is needed to harness the potential of such new means of promoting sustainability education while questions remain around the role of individual curriculum subjects, such as geography, in promoting such transformative learning.

By integrating the study of both natural and social sciences and their interactions, the discipline of geography, which may be considered as "the science for sustainability" has a very long and distinguished history of research and teaching that sees the human–environment relationship as the core of the discipline, and may have an advantage over other disciplines in the facilitation of ESD [8,9]. The concept of sustainable development is embedded in the discipline such that science education and geography exhibit a strong affinity for ESD, which now enjoys wider and deeper implementation in school geography as compared to other subjects [10]. Given the integrative nature of the subject, it seems reasonable to consider geography an appropriate subject for sustainability education [11], which has increasingly become a research focus in the discipline [12]. The evolution of a strong sustainability thread in education means that geography should not merely survive as part of the broader school curriculum but benefit from and, indeed thrive in, the changing educational landscape [13].

Academic research on ESD has been growing, with a body of work spread across the existing domains of environmental education and sustainable development education. In order to assess the relevance of geography in the context of the emerging sustainability discourse, it is necessary to explore the position in the curriculum of key geographic concepts that may be considered instrumental to understanding and addressing sustainability challenges. Nevertheless, there have been very few systematic studies of ESD in school geography courses. This paper explores the role of the school geography curriculum in China and the US in developing learners' awareness, attitudes, and values of sustainable development, benchmarked against a recognized framework for sustainable development competencies as measured against a number of indicators. In order to assess the contribution that geography education in China makes to the promotion of sustainable development, we reflect on four key competencies proposed in the new geography curriculum in China [14], viz. the concepts of human–environment relationships, holistic thinking, regional cognition, and geographical praxis.

Curriculum standards are the criteria underpinning curriculum development, curriculum implementation, courses, and management, which are the most significant central components of a country's educational system [15], stipulating the curriculum concepts, goals, content, implementation procedures, and evaluation methods [16]. In arguing that the promotion of sustainability should also be an essential element of geography, we analyzed the function of learning objectives in China and the National Geography Standards in the United States for high schools and middle schools in cultivating students' sustainability literacy, guided by the following research questions:

RQ1: What is the role of geographic topics in sustainable development?

RQ2: Do the learning objectives in China's and the US's geography curricula reflect the content of sustainable development education?

RQ3: What are the differences in the sustainable development education content reflected in China's middle school geography curriculum and high school geography curriculum?

## 2. Literature Review on Geography Education and ESD

In 2007, the Lucerne Declaration on Geographical Education for Sustainable Development, formulated by the International Geographical Union, was signed in order to emphasize the significance, from a geographic perspective, of ESD [17]. Many countries around the world, such as America, Indonesia, Germany, and Canada, have incorporated the content of sustainable development into primary and secondary school curricula by providing independent courses on sustainable development, integrating ESD into the teaching of other disciplines, and carrying out non-formal education and extracurricular activities regarding ESD. Bednarz sees an important core of geography in "man–land, human–environment, or environment–society relationships" [18], as an appropriate discipline for environmental or sustainability education and research. The topic of geography education has contained the main elements of sustainability, mostly in the field of environmental issues, as well as developing students' knowledge, attitudes, and skills in promoting economic and social sustainability [19]. Bardsley summarized the issues of social and ecological sustainability in geography, which focus on values, skills, and knowledge [20]. Geography education in Germany focuses on different conceptual understandings of the principles of sustainability, mostly in the field of "Human Geography" [21]. In the United States, geography and education for sustainable development are well conceptualized [22], so we chose this developed world geography curriculum as appropriate for an international comparison. Although every state has adopted geography curriculum standards based on Geography for Life, the requirements for geography teaching in the United States vary from state to state and vary greatly within states [23]. As Zadrozny pointed out, in the middle school stage, geography is listed as a branch, and is only one of the knowledge and skills in many subjects. There is no unified curriculum standard in each state [24]. Zadrozny's research also shows that only three states have formulated separate standards related to specific geography courses for grades 6–8 in the NEAP geography assessment by 2018. The teaching of geography courses in other states belongs to a branch of the field of social research or a branch is combined with the geography Curriculum Standards [25]. The National Geography Standards in the United States contain six essential elements and 18 standards, including the ecological perspective in the curriculum aims (Table 1). The fifth element, "Environment and Society," reintegrates the content of the geography curriculum by emphasizing the interactions between material and human systems and establishing the centrality of resources in the "environment–society" relationship [26].

**Table 1.** "Environment and Society" in the National Geography Standards in the USA.

| Stage | Issue Time | Curriculum Aims |
|---|---|---|
| K-12 | 2012 [26] | Understanding and using spatial and ecological perspectives helps geographers understand how to interpret nature and societies on Earth. Viewed together, the geographic perspective overall encompasses an understanding of spatial patterns and processes on Earth and its web of living and nonliving elements interacting in complex webs of relationships within nature and between nature and societies. |

In China, students are required to follow the uniform national curriculum standards, and geography education, which spans from grades 1 to 12 and is integrated with science and social studies since 1988. In the elementary stage, geographical ideas, such as the relationship between human activities and the geographical environment, and the basic national policies related to population, resources, and the environment are incorporated into social studies. When students enter middle school (grades 7–9), geography is a mandatory subject required by China's nine-year compulsory education. At the high school level

(grades 10 to 12), geography is one of six optional subjects: politics, history, geography, physics, chemistry, and biology. The curriculum standards for geography, developed by China's Ministry of Education, is the core document guiding grades 6–12 geography education nationwide and has gone through seven curriculum reforms so as to improve the quality of basic education, expand students' core strengths, and thus facilitate ESD. In 2001, the Full-time Compulsory Education Geography Curriculum Standard (Experimental Draft) [27] was issued, which includes the following four units: The Earth and Maps, World Geography, China Geography, and Local Geography. It mainly emphasized the two characteristics of the Geography curriculum: comprehensiveness and regionality, but it has not elaborated on sustainable development. Since 2001, geography has undergone a major paradigm shift. Middle school Geography Curriculum Standards were revised three times and high school Geography Curriculum Standards were revised twice. The Compulsory Education Geography Curriculum Standard (2011 Edition) was published in 2011, which specifically pointed out that the "geography curriculum highlights the population, resources, environment, and development problems faced by today's society, and expounds the scientific concept of population, resources, environment, and sustainable development", which increases the attention towards sustainable development. The General High School Geography Curriculum Standard (2017 Edition) was revised again in 2017, arising from which key competencies of the subject were proposed for the first time [14], including the concept of human–environment relationships, holistic thinking, regional cognition, and geographical praxis. In high school courses, the geography curriculum has an important focus on sustainability, which emphasizes students' understanding of sustainable development in relation to the concepts of population, resources, the environment, and society.

According to one of the key competencies evident in China's geography curriculum at middle school and high school, the concept of human–earth coordination is regarded as important and aims at encouraging learners to understand and respect the relationship between human activities and the environment. Through geography learning, students can understand geographical issues and look at the relationship between people and the earth from the perspective of both physical and socio-cultural elements. Human activities clearly have a profound impact on the environment, and this competency focuses on understanding the relationship between people and nature dialectically. In addition, students should be able to actively explore the human–land relationship problems in practice and develop appropriate value judgment skills. The course objectives are determined according to the key competencies and contents (Table 2).

**Table 2.** The related expression of "man–earth coordination view" in the Geography Curriculum Standards of middle school and high school in China.

| Stage | Issue Time | Curriculum Aims |
|---|---|---|
| Middle School Geography Curriculum Standards | 2011 [28] | Understand the major issues of population, resources, the environment, and development faced by humankind, and develop a preliminary understanding of the relationship between the environment and human activities. To "enhance the awareness of the rule of law in protecting the environment and resources, preliminary form the concept of sustainable development, and gradually develop the habit of caring for and caring for the environment". |
| Middle School Geography Curriculum Standards | 2022 [29] | Students can preliminarily understand that the geographical environment is the basis for human survival, that human activities have a profound impact on this environment, and that coordinating the relationship between people and nature is necessary for the sustainable development of human society. With the aim of improving students' key competence, geography courses guide students to learn geography that proves useful for life and lifelong development, which lays a solid foundation for cultivating next generations with the idea of ecological civilization. |

**Table 2.** *Cont.*

| Stage | Issue Time | Curriculum Aims |
|---|---|---|
| High School Geography Curriculum Standards | 2003 [27] | Environmental education and sustainable development are highlighted "to establish the concept of sustainable development, to form a civilized way of life and production, to enhance students' sense of responsibility, and to strengthen the concept of sustainable development in which population, resources, environment, and society coordinate with each other". There is a focus on population, resources, the environment, regional development, and other issues, so as to help students to correctly understand the relationship between man and land, form thoughts on sustainable development, care for the Earth, and treat the environment well. The current situation and trend of China's environment and development are highlighted, and an understanding of the global environment and development issues is fostered, thereby enhancing the social responsibility of caring for the environment. |
| High School Geography Curriculum Standards | 2017 [14] | Develops awareness of local, national, and global geographic problems and sustainable development issues. Recognize the importance of human–land coordination for sustainable development, and form the value of respecting nature and harmonious development. |

## 3. A Framework for ESD Competency Analysis

Education is key to improving students' abilities to understand and potentially resolve sustainability issues. The United Nations Conference on Environment and Development in Rio de Janeiro in 1992 established the three basic dimensions of sustainable development, *viz.* the ecological, economic, and social dimensions [30], and there is a strongly held view that these should all be incorporated into school education [31]. ESD generally focuses on the development and strengthening of individual competencies, with the goal of enabling individual learners to contribute to and participate in a range of sustainable development actions. The international consensus has been reached regarding a definitional framework for ESD literacy [32] that targets the development of attitudes, skills, perspectives, and knowledge in relation to the environment and that helps citizens of the world make more informed decisions and act upon them in such a way as to develop a more sustainable future [33].

ESD in China is aimed at promoting the sustainable development of society, the economy, the environment, and culture. Item 62 of the Education 2030 Framework for Action [34] states that ESD and Global Civic Education [35] (GCED) incorporate peace and human rights education, as well as promoting intercultural education and the achievement of international understanding so that citizens have access to the necessary knowledge, skills, values, and insights to lead a full and substantial life, make informed decisions, and actively assume responsibility for addressing local and global challenges.

A framework for the definition of sustainability literacy (see Table 3) has been developed based on two important frameworks in particular, societal, economic, environmental, and cultural dimensions (UNESCO [36] and the China ESD Roadmap [37]). In this study, we employ a framework for sustainable development literacy consisting of four domains: (1) knowledge, (2) competencies, (3) attitudes and values, and (4) actions and behaviors, covering elements common to the various definitions and frameworks of sustainable development literacy. Sustainable development knowledge refers to an understanding of the scientific basis of sustainable development across the social, economic, environmental, and cultural domains. Sustainable development competencies include: (i) the ability to collect and process information, (ii) accurate and organized expression, (iii) the ability to evaluate other people's opinions and draw conclusions, (iv) teamwork and coordination, (v) the ability to solve practical problems of sustainable development, (vi) systemic thinking, (vii) critical thinking, and (viii) predictive thinking. Sustainable development attitudes and values refer to (i) the respect for present and future generations, (ii) the respect for differences and diversity, (iii) the respect for the environment, and (iv) the respect

for resources. Sustainable development actions and behaviors refer to green diets, green lifestyles, eco-tourism, and low carbon consumption to cultivate sustainable lifestyles.

**Table 3.** ESD literacy framework.

| Dimension | Code | Items |
|---|---|---|
| Knowledge | A | Knowledge of social sustainability |
| | B | Knowledge of economic sustainability |
| | C | Knowledge of environmental sustainability |
| | D | Knowledge of cultural sustainability |
| Competencies | E | The ability to collect and process information |
| | F | Accurate and organized expression ability |
| | G | The ability to evaluate other people's opinions and book conclusions |
| | H | Teamwork and coordination ability |
| | I | The ability to solve practical problems of sustainable development |
| | J | Systemic thinking |
| | K | Critical Thinking |
| | L | Predictive thinking |
| Attitudes and values | M | Respect for present and future generations |
| | N | Respect for differences and diversity |
| | O | Respect for the environment |
| | P | Respect for resources |
| Actions and behaviors | Q | Green diet, green lifestyle, eco-tourism, low carbon consumption |

## 4. Materials and Methods

### 4.1. Research Design

This study deployed content analysis, widely applied in the field of social sciences and humanities [38] as a qualitative method. China's and the US's curriculum standards for geography in middle school and high school were analyzed as data sources. The discipline of geography is afforded particular representation and importance, which reflects the level of integration and development of sustainable education in compulsory education, including basic knowledge and skills. This approach provides a unique opportunity to systematically analyze the learning objectives in China's and US's curriculum standards for geography (see Table 4), of which there are 239 learning objectives in China (98 items in middle school and 141 items in high school) and 165 learning objectives in the US (80 items in middle school and 85 items in high school) that allow us to perform a thorough content analysis.

**Table 4.** Learning objectives in geography curriculum standards in China and America.

| | China | America |
|---|---|---|
| Middle school geography | 98 items | 80 items |
| High school geography | 141 items | 85 items |
| Total | 239 items | 165 items |

First, we developed a definition framework for ESD literacy and then used this as a coding scheme for content analysis. Thereafter, we applied cluster analysis, a multivariate statistical technique that organizes information based on similarities or dissimilarities and the location of words in relation to other words or co-occurrence. This type of analysis is typically visualized in a dendrogram, a branching diagram that represents a hierarchy of categories based on the number of shared characteristics [16]. The WordStat 8.0 program is used to conduct second-order clustering based on the similarity of terms that link words that are semantically related. Jaccard's coefficient was then deployed to reveal the degree of similarity among the words within the dendrograms [39]. The chosen number

of clusters then identifies how the words are associated and separated within the context in question [40]. Moreover, link analysis is used to represent the connection between keywords or dictionary items visualized through the network diagram [39], which allows one to visualize the connections between keywords or dictionary items using a network graph, the thickness of the line representing the strength of this relationship [41].

### 4.2. Analysis of Encoding Process and Efficacy

China's geography curriculum standards (see Table 5) for grades 7 to 9 include the following four units: The Earth and Maps, World Geography, Geography of China, and Regional Geography. The equivalent for high school (grades 10 to 12) includes five units: Geography 1, Geography 2, Foundation of Physical Geography, Regional Development, Resources, Environment, and National Security (see Table 6).

**Table 5.** Environmental related concepts—topics in the geography curriculum in China.

| Grade | Unit | | |
|---|---|---|---|
| **World Geography** | | | |
| 7th grade | Using maps and data, identifying the proportion of sea and land on the surface of the earth, and describing the distribution characteristics of sea and land. | Using maps and other materials, pointing out one or several natural resources that have a greater impact on local or world economic development, and identifying their distribution, production, and export. | Identifying the particularities of the natural environment in the South and Arctic regions, and understanding the importance of carrying out polar scientific investigations and protecting the polar environment. |
| **Geography of China** | | | |
| 8th grade | Giving an example of the difference between renewable and non-renewable resources. | | |

Source: Compulsory Education Geography Curriculum Standards (2011) [28].

**Table 6.** Environmental related concepts—topics in the geography curriculum in China.

| Grade | Unit | | |
|---|---|---|---|
| | Compulsory Course | Compulsory Course | |
| | Geography 1 | Geography 2 | |
| 10th grade | Identifying the main vegetation through field observations or using videos and images, and explaining its relationship with the natural environment. | Using data to describe the main environmental problems facing humanity and explaining the main ways and reasons for coordinating human–environment relations and sustainable development. | |
| | Optional Compulsory Course 1 | Optional Compulsory Course 2 | Optional Compulsory Course 3 |
| | Foundation of Physical Geography | Regional Development | Resources, Environment and National Security |
| 11th grade | Using diagrams to analyze the impact of air–sea interactions on the global water and heat balance, and explaining the impact of El Niño and La Niña on the global climate and human activities. | Taking an ecologically fragile area as an example to illustrate the environmental and development problems in this type of area, as well as comprehensive treatment measures. | Combining examples to explain the significance of setting up nature reserves to ecological security. |

Source: General high school geography curriculum standards (2017) [14].

The National Geography Standards in the United States include six essential elements (see Table 7): The World in Spatial Terms, Places and Regions, Physical Systems, Human Systems, Environment and Society, The Uses of Geography.

**Table 7.** Environment and Society in the geography curriculum in America.

| Grade | Geography Standard 14: How Human Actions Modify the Physical Environment | | |
|---|---|---|---|
| | Modification of the physical environment. | The use of technology. | Consequences for people and environments. |
| 8th grade | Describe and explain how human-induced changes in one place can affect the physical environment in other places. | Describe and explain the ways in which technology has expanded the scale of human modification of the physical environment. | Analyze the positive and negative consequences of humans changing the physical environment. |
| 12th grade | Explain the global impacts of human changes in the physical environment. | Evaluate the intended and unintended impacts of using technology to modify the physical environment. | Describe and evaluate scenarios for mitigating and/or adapting to environmental changes caused by human modifications. |
| | **Geography Standard 15: How physical systems affect human systems** | | |
| | Environmental opportunities and constraints | Environmental hazards | Adaptation to the environment |
| 8th grade | Explain how the characteristics of different physical environments offer opportunities for human activities. B. Explain how the characteristics of different physical environments place constraints on human activities. | Describe and explain the types and characteristics of hazards. Explain the causes and locations of various types of environmental hazards. | Explain how people use tools and technologies in adapting to the physical environment. |
| 12th grade | Explain how people may view the physical environment as either an opportunity or a constraint depending on their choice of activities. | Explain and compare how people in different environments think about and respond to environmental hazards. Explain how environmental hazards affect human systems and why people may have different ways of reacting to them. | Explain how societies adapt to reduced capacity in the physical environment. Analyze the concept of "limits to growth" to explain adaptation strategies in response to the restrictions imposed on human systems by physical systems. |

Source: Geography for Life: National Geography Standards, Second Edition (2012) [42].

The 239 learning objectives in China's geography curriculum standards and 165 learning objectives in American geography curriculum standards were assigned according to the coding framework obtained from the Sustainable Development Goals 4 [43], after multiple trials. Based on the conceptual definition of the primary and secondary indicators, two coders counted the frequency of occurrence of specific examples (including sentences, long phrases, and clauses).

To establish the reliability and validity of the coding process, the two coders independently coded a sample of randomly chosen learning objectives (20%) and then compared their coding results and discussed their initial coding strategies. Both the percentage agreement (98.8%; 80% for coding the random samples) and the kappa value (0.863; calculated with SPSS 23.0), which were calculated before the final consensus was reached, suggest the coding results are reliable and valid for use in the study. After confirming this reliability, the coding discrepancy between the two coders was assessed to ensure that the final coding was consistent according to the coding schemes and coding strategies.

### 4.3. Establishment of Category Terms

The terms were fashioned based on a thorough review of the literature as well as proximity plots and cluster analysis of the text. This entailed multivariate analysis techniques that organize information based on similarities or dissimilarities and location of

words in relation to other words or co-occurrence surrounding the terms values, sustainability, and responsibility (for texts that did not contain the words values or sustainability). Profiling the documents addresses how the dominant discourse, general categories, and patterns regarding ecological sustainability and environmental values are expressed in each curriculum standard.

This phase of analysis began with an examination of the top word frequencies in each of the texts. The determination of high word frequencies for each document also assisted in creating general profiles.

Terms were categorized, reshuffled into different groups, and renamed. The following categories of terms were established: general education, Science education processes, ecology, environmental education, environmental agency, sustainability terms, and geography terms. In this study, each of the geography curriculum standards was analyzed to see how high-frequency words tended to be grouped within the categories.

## 5. Results

The results of this study indicate that the geography curriculum has played an important role in developing learners' ESD literacy. Many learning objectives in China's and the U.S.A.'s geography curricula reflect the role of the geography curriculum in cultivating learner's ESD literacy.

### 5.1. The Role of China's Geography Curriculum for ESD

Profiling the documents addresses question one regarding how the geography curriculum in China contributes to ESD. The geography curriculum standards were analyzed to see how high-frequency words tended to group within the categories. The overall theme in China's geography curriculum standards lies in geography, the environment, and cognitive skills. An examination of the most frequent words shows, as follows, the most frequent words in China's middle school and high school geography curriculum standards. As noted in Figure 1, the top five words in China's geography curriculum are: illustrate, development, state, geographic, and analyze. The word "development" (51 times), geographic" (28 times), "maps" (25 times) and "sustainable" (10 times) speak to the focus of geography education and ESD. The words "illustrate" (60 times), "state" (29 times), "analyze" (25 times), "describe" (24 times), "understand" (21 times), and "summarize" (13 times) are cognitive skills terms and science-teaching terms. The words "environment" (21 times), "resources" (20 times), "water" (17 times), "climate" (15 times), "earth" (15 times), "disasters" (14 times), "economic" (14 times), "land" (14 times) are ecological and environmental education terms. These keywords lie within the category of general education and geography education, which aligns with the initial impressions from early readings of the documents that the geography curriculum in China has positive effects on ESD.

**Figure 1.** Distribution of keywords in Chinese Geography Curriculum Standards.

After classifying these keywords into terms (as is shown in Table 8), the term categories reveal differences in the geography curriculum standards for middle school and high schools in China. Middle school geography curriculum standards are mainly composed of ecology terms and Science education process terms, while high school geography curriculum standards mainly involve sustainability eco-values, life-supporting, broad education, and environmental terms. In general, these are ecological and environmental education terms, geography terms, environment, and cognitive skills terms. These high-frequency terms are scattered across many categories and all relate to the concept of a sustainable future.

**Table 8.** A summary of the document profiles in China.

| Stage | Category | Overall Description |
|---|---|---|
| Middle school | Ecology; Science education processes | List of science topics; Attitude and values terms absent |
| High school | Sustainability eco-values; Life-supporting General education; Environmental terms | Call to action, each statement begins with a verb Issue analysis; Multidisciplinary; Inclusive of values |
| Overall | Geography terms; Environment and cognitive skills | Focus on pedagogy; Inclusive of values |

Cluster analysis (as shown in Figure 2) was applied to examine the relatedness of all the geography curriculum standards as a whole. The dendrogram (Figure 2) was formed using the crosstabs feature. Similarity decreases from left to right, which yields five main clusters, viz.: Group 1 comprises nine words, from "describe" to "illustrate"; group 2: "economic"; group 3: "state" and "summarize"; group 4: "development"; group 5: "sustainable". The findings align with the pie graph. The dendrogram clearly shows that middle school and high school geography education in China has a strong focus on the environment, resources, analytical ability, and sustainable development. According to the results, the specific indicators of sustainable development knowledge are summarized as follows: (1) "Knowledge of Social Sustainability" includes: population issues, government and the rule of law, poverty issues, globalization issues, and international cooperation; (2) "Knowledge of Economic Sustainability" includes: eco-city construction, industrial location, science and technology, and economic growth; (3) " Knowledge of Environmental sustainability" includes the human–environment relationship, ecosystem, natural resources (water resources, soil resources, and renewable resources), atmosphere, environmental pollution, global warming, resources, the environment, and protection issues, and climate change; (4) "Knowledge of Cultural Sustainability" includes: cultural diversity, tourism, regional culture, regional culture, agricultural culture, cultural landscape, and cultural heritage.

*5.2. Learning Objectives That Reflect the Content of ESD in China*

China's geography curriculum standards also reflect the cognitive skills dimension of the teaching goals and learning objectives and highlight "the ability to improve the geography of sustainable development" in the Declaration of Geographic Sustainable Development [44], which is the most important geographical ability to implement sustainable development. The content analysis of learning objectives reveals that the middle and high school geography curriculum in China has the potential to significantly cultivate students' sustainable development literacy, especially in terms of "environmental sustainable development knowledge" and "sustainable development competencies", which are presented in Table 9. There are 161 learning objectives that contain knowledge that may be coded as ESD concepts, principles, laws, theories, and models. As noted above, each learning objective

indicates specific demands for students regarding both knowledge and cognition, that is, what and how students need to learn. Around half of the knowledge that students were supposed to learn in their geography teaching was coded as belonging to the knowledge domain and pertained to Earth science and physical geography. The indicator "systemic thinking" is counted 238 times, while "accurate and organized expression" received 168 coding references. "The ability to collect and process information" and "critical thinking" in the geography curriculum standards are coded 94 times and 64 times, respectively, which aligns with the competencies learners are expected to achieve. Table 9 also shows that the sustainable attitudes and values domain in ESD competencies was only slightly represented in China's geography curriculum. It also reveals that the Geographic curriculum standards pay little attention to the sustainable behavior and action dimensions. It appears, then, that students' competencies for sustainable development are highly emphasized in China's geography curriculum, followed by environmental knowledge for sustainable development. Of less obvious importance, however, are attitudes and values, and sustainable behaviors and actions appear to be largely neglected in China's geography curriculum.

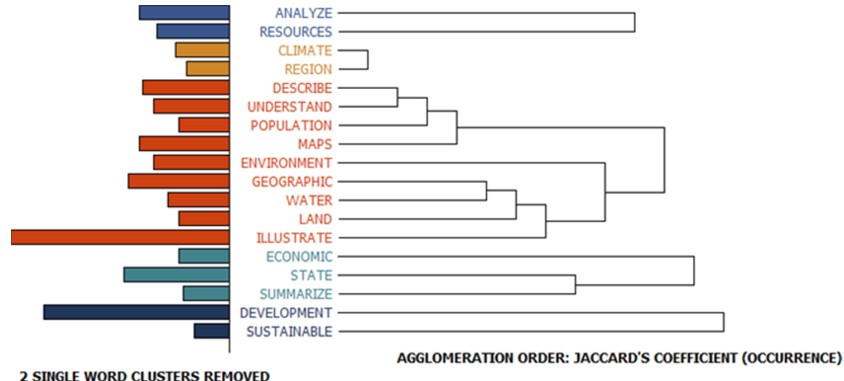

**Figure 2.** Dendrogram of Chinese Geography curriculum standards.

**Table 9.** Results of the ESD literacy by content analysis in China.

| Sustainable Development Literacy | Specific Content | Frequency | Percentage | Chi-Square Value |
|---|---|---|---|---|
| Knowledge | A. Knowledge of social sustainability | 25 | 2.98 | 0.500 |
| | B. Knowledge of economic sustainability | 32 | 3.81 | |
| | C. Knowledge of environmental sustainability | 161 | 19.17 | |
| | D. Knowledge of cultural sustainability | 12 | 1.43 | |
| Competencies | E. The ability to collect and process information | 94 | 11.19 | 0.750 |
| | F. Accurate and organized expression | 168 | 20 | |
| | G. Ability to evaluate other people's opinions and book conclusions | 2 | 0.24 | |
| | H. Teamwork and coordination | 0 | 0 | |
| | I. Ability to solve practical problems of sustainable development | 26 | 3.10 | |
| | J. Systemic thinking | 238 | 28.33 | |
| | K. Critical Thinking | 64 | 7.62 | |
| | L. Predictive thinking | 6 | 0.71 | |
| Attitudes and values | M. Respect for present and future generations | 1 | 0.12 | 0.500 |
| | N. Respect for differences and diversity | 1 | 0.12 | |
| | O. Respect for the environment | 7 | 0.83 | |
| | P. Respect for resources | 3 | 0.36 | |
| Actions and behaviors | Q. Green diet, green lifestyle, eco-tourism, low carbon consumption | 0 | 0 | constant |

Note: The chi-square statistics are test degree of evenness of proportions across different categories of each variable. Differences are statistically significant at the 5% level of significance. Source: geography curriculum standard collected by the authors.

China's geography curriculum standards mainly reflect the dimension of "ecological integrity", and pay great attention to teaching methods. They pay more attention to the ecological environment and biodiversity in the theme of the concept of sustainable development, and respect and care for all living things, from which it can be concluded that China's geography curriculum standards focus on both the knowledge of environmental sustainable development and the competencies of sustainable development.

### 5.3. Learning Objectives That Reflect the Content of ESD in America

The National Geography Standards in the United States outline the topics of learning objectives for each grade level, including ecology, environmental literacy, energy, and biogeochemical cycles in the Earth system, and skills relating to geographical investigation and experiment are also emphasized. The most commonly appearing words are related to ecology, including energy, Earth, carbon, and ecosystems, such as agriculture, health, limited resources, or pollution. The National Geography Standards in the United States showed great emphasis on human beings, the Earth, and the environment, which indicates the importance of a harmonious human–environment relationship on the ecosystem. As is shown in Figure 3, the distribution of keywords clearly shows that American geography curriculum standards focus a lot on the Earth, humans, the environment, resources, sustainability, and global and social development issues.

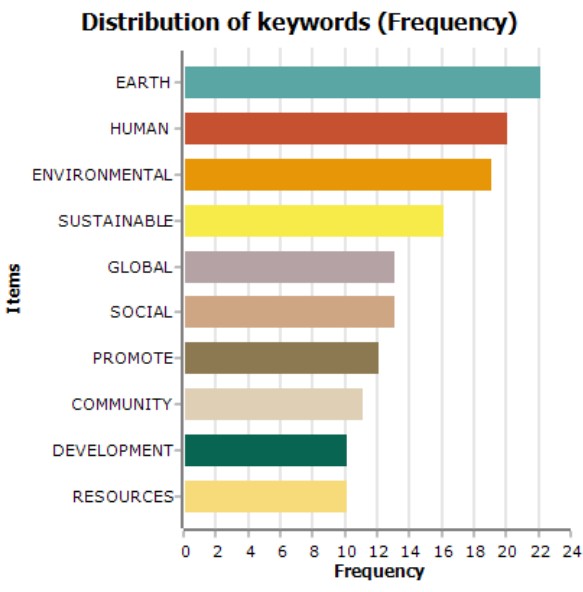

**Figure 3.** Distribution of keywords in American geography curriculum standards.

The National Geography Standards in the United States (as shown in Figure 4) show that the most relevant topics in relation to sustainable development include "humanity", "globalization", "community", "environment", "Earth", "resources", and "society". The American geography curriculum standards focus on the reasoning of geographical knowledge and geographical phenomena, such as one of the learning objectives "explain the distribution process of human and natural phenomena", which indicates that most of the sustainability expressed in the American geography curriculum standards is related to human beings, the Earth, and the environment.

As is shown in the link analysis graph (Figure 5), there is a strong correlation between "global" and "sustainable", and a high level of interactivity among "human", "environment", "sustainable" and "development". Such a relationship is represented as nodes connected by a line, and the thickness of the line indicates the strength of this relationship. As is shown, "human" and "sustainable" have the thickest connecting line, which indicates the strongest correlation.

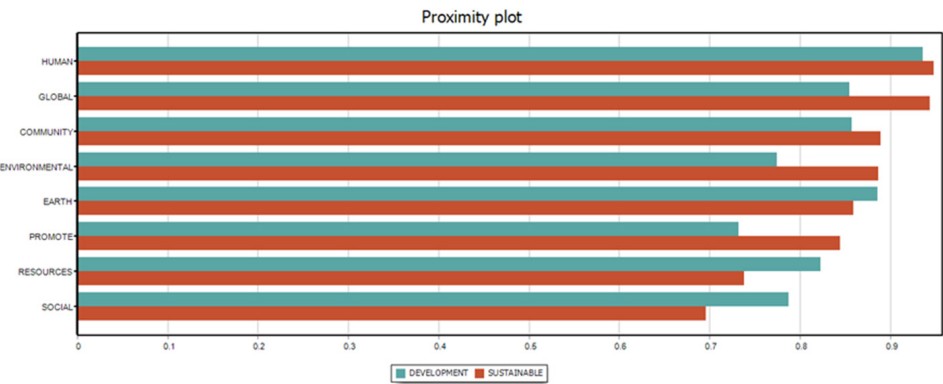

**Figure 4.** Proximity plot of American geography curriculum standards (Jaccard's coefficient similarity unit).

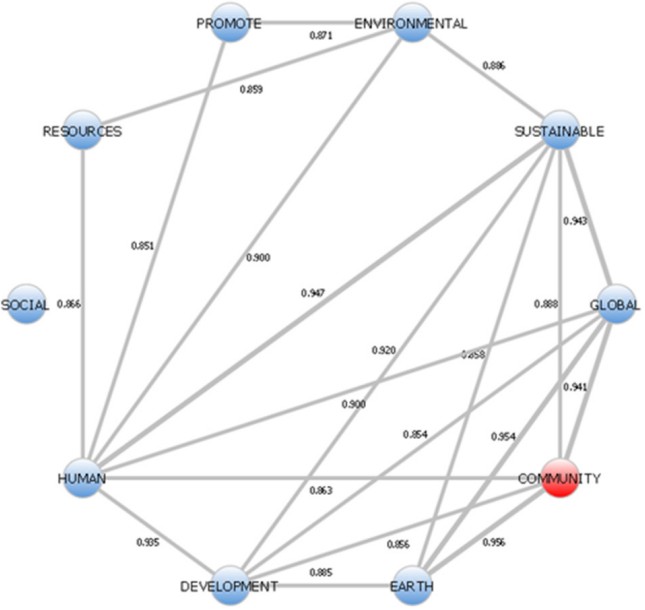

**Figure 5.** Occurrences link analysis of American geography curriculum standards.

*5.4. Differences in the Content of Middle School and High School in China*

The difference in content between China's middle school geography curriculum and high school geography curriculum was also explored. As shown in Figures 6 and 7, China's middle school geography curriculum focuses more on environmental and geographical topics, while the high school geography curriculum focuses a lot on ESD themes. In the data analysis, the frequency of "illustrate", "country", and "characteristics" received the most coding references, which reveals that the middle school geography curriculum focuses on geographical and developmental themes. The words "natural", "climate", "geographic", "impact", "economic", "population", "world", "environment", "hometown", "region", and "resources" are the environmental and geographical terms.

According to the geography curriculum standards in China, the content of sustainable development education involved in the geography curriculum is very extensive, with a core focus on understanding the interdependent and interrelated connections between humans and the environment. As the curriculum objectives of compulsory education (2022) indicate, students are required to develop a preliminary understanding that the geographical environment is the basis for human survival, human activities have a profound impact on the geographical environment, and coordinating the relationship between people and nature is essential for the sustainable development of human society [29].

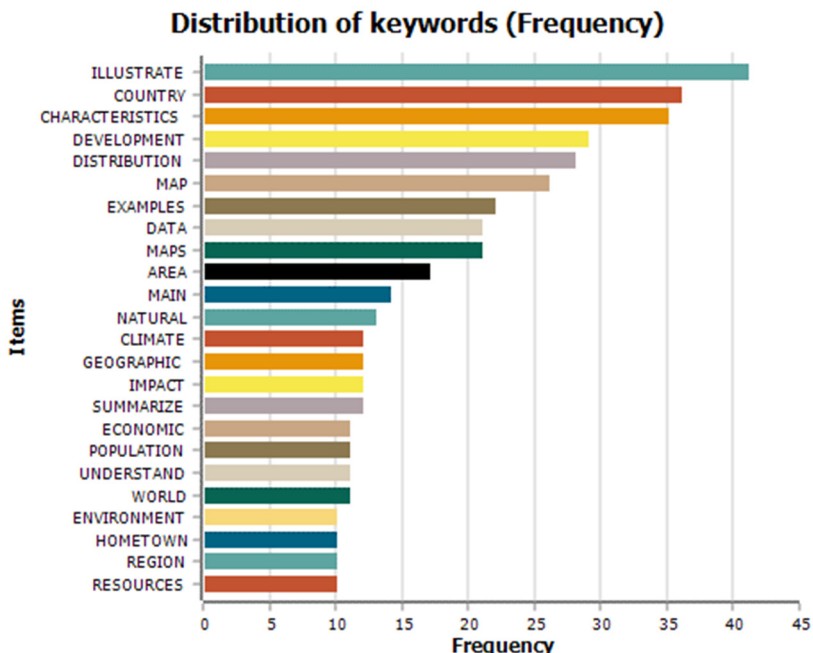

**Figure 6.** Distribution of keywords in Chinese middle school.

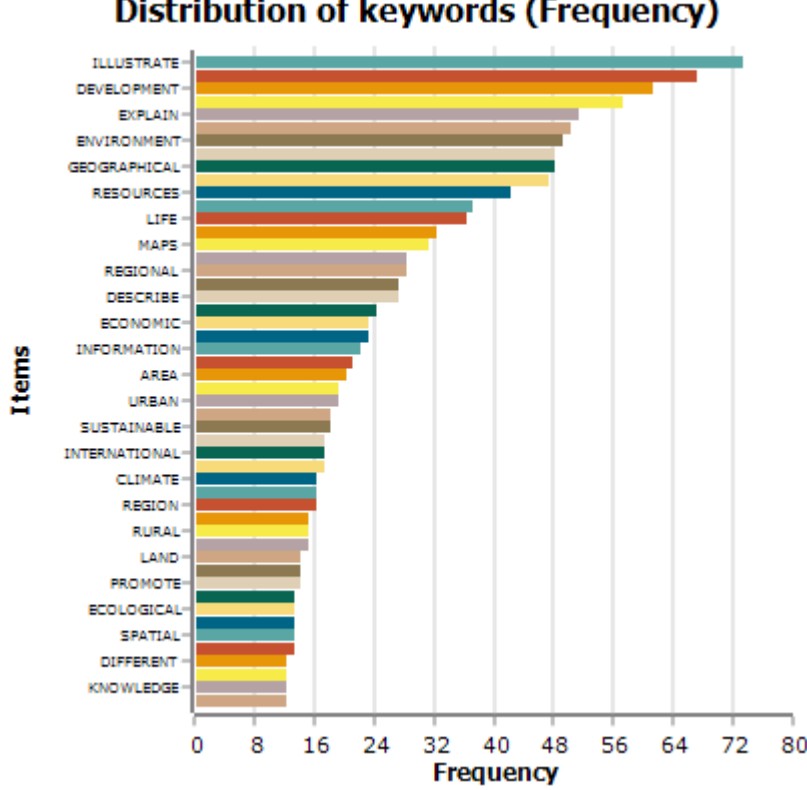

**Figure 7.** Distribution of keywords in Chinese high school.

As shown in Figure 7, the most frequent keywords in China's high school geography curriculum are "illustrate" and "explain", which also reveals that high school geography focuses on geographical and developmental themes. The words "resources", "methods", "diagram", "earth", "global", "regional", "environmental", and "sustainable" are environmental and sustainability terms. This shows that the high school geography curriculum focuses more on sustainable topics than the middle school geography curriculum. The

content of sustainable development education involved in high school geography mainly includes: the impact of the natural environment on human activities, the coordinated development of human and geographical environment, regional sustainable development, and so on. The content of sustainable development education required to be mastered in the geography course can promote students' mastery of the methods to solve the problems of humans, nature, and social sustainable development; cultivate students' practical ability to participate in sustainable development; develop responsible emotional attitudes and values towards nature and society; and improve students' awareness and behavior towards participation in sustainable development and human sustainable development.

The results above align with the high school geography curriculum objectives in China. Students are expected to be able to view the interaction between the geographical environment and human activities; deeply understand the different ways, intensities, and consequences of their interaction; understand the phased performance of people's understanding of the relationship between humans and land and its causes; recognize the importance of human land coordination for sustainable development; and form an attitude of respecting nature and harmonious development [14].

## 6. Discussion

In order to investigate whether, to what extent, and on what aspects the geography curriculum has the potential to contribute to the facilitation of students' ESD literacy, we examined China's and the US's middle school and high school geography curriculum standards from the perspective of ESD literacy. The results of the present study contribute to our understanding of the presence of the knowledge and competencies of sustainability in geography education in China and the U.S.A. The findings are aligned with one of the curriculum objectives of the geography curriculum standard, which highlights the inter-active relationship of "science, technology, society, and the environment" [14]. Applying the method of content analysis with the support of the software WordStat 8.0, we came to the following results: The learning objectives of the geography curricula from middle school to high school were closely scrutinized for their emphasis given on sustainability. In our analysis of the curriculum standards, we found that the geography curriculum, besides containing some ESD knowledge (i.e., knowledge about physical geography and Earth science and the environment), had a potential contribution to developing students' ESD literacy, especially in terms of the concepts of "human–environment coordination", "accurate and organized expression", and "systemic thinking".

The findings of this study indicate that China's geography curriculum and the U.S.A.'s geography curriculum both focus on the human–environment relationship, which focus on human–land relationships and interactions between human activities (industrial production and agricultural production) and the geographical environment. These findings reinforce the scope of geography studies, covering social geography and physical geography [45]. Chinese human geography upholds the subject's designation as being integrated and inter-disciplinary. Research focuses on interactions between the natural and human spheres of the Earth's surface, and it is guided by the understanding and effects of the processes of regional sustainable development at different spatial scales [46]. The U.S.A.'s geography Curriculum focuses on geographic perspectives, cultivating students' ecological view of the world from multiple perspectives, focusing on the relationships between human societies and ecosystems, and viewing the world as a network of relationships between living things and nonliving factors [26]. The concept of human–earth coordination is one of the core concepts of modern geography education. From the perspective of the human–earth relationship, cultivating students' understanding of the impact of the geographical environment on human beings, the impact of human activities on the geographical environment, and the coordination of the man–Earth relationship. The relationship between people is also the core content in geographical research. Coordinating the relationship between human activities and the environment is required to establish a harmonious relationship between humans and nature [29]. The cultivation of the concept of human–earth coordination helps

students form the concepts of respecting and protecting nature and green development, nourishes humanistic feelings, and enhances their sense of social responsibility.

There are many similarities and parallels between the geography curricula in China and the US, as both do seem to address sustainable development issues in some detail. Overall, China focuses more on knowledge and learning methods, such as systems thinking, while the US American geography curriculum appears to focus more on "doing geography", fieldwork, and geographical skills, including GIS. People with geographical literacy are able to use geographic skills to investigate, identify, and solve geographic problems.

From the perspective of curriculum, the promotion of sustainable development education should focus on the teaching and learning methods of sustainable development education. Education for sustainability means the creation of space for transformative social learning [47], which emphasizes "learning for being", alongside learning for knowing and learning for doing [48]. Such as "climate change" as an issue of conflicting conceptualizations at a time when there is little room for learning in the formal curriculum, and the assessment is of no value if learning does not translate into action [49]. Systemic thinking has long been recognized as a central component of achieving sustainability competencies [50] and, most recently, was identified as a widely recognized key competency in sustainability education [51]. With the need for learners to be able to understand systems and see the world as an interconnected whole, appreciating the connections between human and natural environments and recognizing the consequences of actions taken and the causes of unsustainability, systemic thinking competency is an important ability to collectively analyze complex systems across different domains (society, the environment, the economy, etc.) and across different scales (local to global), thereby considering cascading effects, inertia, feedback loops, and other systemic features related to sustainability issues and problem-solving learning [51].

Also of importance is the capacity to recognize the values underlying the actions of individuals in developing ESD [52]. Human behavior is grounded in values, and changes in societal behavior depend on changes in values [53]. The previous study revealed that education can be held partially responsible for both problems and solutions in creating a sustainable future. Orr (1992) points out that sustainability challenges are not attributed to a crisis in education [54], but rather a crisis of education. As an available tool, education is one of the main means to solve the environmental problems of sustainable development [55], which is used to formulate the medium and long-term strategy of critical thinking to change human natural behavior. Critical thinking is also one of the key competencies in ESD, which falls under the cognitive domain. Developing critical thinking is a critical prerequisite for building sustainable citizens. It is vital to "mainstream" ESD in order to develop critical thinking and other key competencies to survive and thrive in an uncertain and technology-driven future. As a person's attitude and behavior towards nature are largely determined by the values they acquire in the process of education [56]. Even though environmental competencies are important steps towards changing environmental behaviors, there are many other influences that affect pro-environmental behaviors. Knowledge is a necessary, but not sufficient, precondition for developing pro-environmental behaviors [57]. A gap exists between knowledge and action in which complex factors are at play [58]. Environmental behavior is determined by a myriad of variables and variable interactions; there is no single variable explanation. According to the New Ecological Paradigm Scale (Dunlap, 2000), environmental worldviews were added to the discussion of what motivates pro-environmental behavior. According to Stern et al., values-beliefs-norms (VBN) is the theory of environmental concern and behavior [59], which could underlie an individual's beliefs and, in turn, affect personal norms; together, these influence behaviors [60]. Therefore, establishing learners' sustainability values in relation to the environment is essential to ESD literacy. ESD values and the integrated, transformative social learning model in approaching ESD supported by the literature are not explicitly addressed if geography education learning objectives are based on the geography curriculum standards' descriptions.



Overall, the geography curriculum in China has transformed from knowledge-centered instruction to the improvement of students' geographical literacy levels by highlighting systemic thinking and the interactive relationship of science, technology, society, and the environment. The US's geography curriculum emphasizes the use of geographic perspectives, content knowledge, and skills to "do geography" as an active inquiry or study [61]. As the curriculum has attached greater importance of ESD literacy to students' overall competencies, students have begun to pay increased attention to the global challenges we face, including population, climate change, environmental degradation, and inequality from a developmental perspective. The purpose of standards for the US's geography curriculum is to bring all students up to internationally competitive levels to meet the demands of a new age and a different world [62]. The concept of "doing geography" in the US's geography curriculum standards strengthens the importance of geographic perspective in cultivating students' literacy, which combines geographical knowledge, skills, and geographical thinking. It also highlights the function of spatial and ecological perspectives, which is similar to the curriculum aims in Chinese geography. It is pointed out in the Chinese geography curriculum standards that students should be "guided to think from the perspective of geography, pay attention to nature and society, so that they can gradually foster the view of "human—land coordination" and sustainable development, which lays the foundation for cultivating geographically literate citizens". Therefore, it is recommended that China should pay attention to the process of geographical inquiry learning in the instructional process, strengthen the cultivation of students' geographic skills to solve authentic issues, and emphasize the application of geographic knowledge through geographical field trips.

## 7. Conclusions

In this paper, we have presented the features of geography and explored the value of geography education in promoting learners' literacy in sustainable development, which indicate that the geography curriculum has the potential to contribute to developing ESD literacy for a range of demonstrated reasons. Our findings suggest that geography education plays an important role in cultivating learners' sustainable development literacy, mainly in the aspect of cultivating students' knowledge and competencies in sustainability. However, this study had limitations in that it did not include the evaluation of the learning outcomes in sustainability. This study also was limited in the geography curriculum in China and the US, which may need to facilitate the integration of learning methods with ESD. The results of this research have also identified a challenge to integrate ESD across the middle school and high school geography curricula. Overall, the findings of this study can help policymakers, researchers, curriculum developers, and teachers to develop a clear understanding of the existing curriculum content and to take steps towards improving it to align with the principles of sustainability.

**Author Contributions:** Conceptualization was performed by S.M. and Y.D.; S.M. contributed to the methodology, software, and investigation, as well as the preparation of the original draft; S.M. contributed to the resources and data curation, as well as visualizations and project administration; M.E.M. and Y.D. contributed to the conceptualization, review and editing of the manuscript, supervision of the project, and funding acquisition; F.G. contributed to the methodology and resources. All authors have read and agreed to the published version of the manuscript.

**Funding:** This research was funded by The National Social Science Fund of China's 'the Twelfth Five-year Plan', grant number is AHA120008.

**Institutional Review Board Statement:** This study did not require ethical approval.

**Informed Consent Statement:** This study did not involve humans.

**Data Availability Statement:** This study did not report any data.

**Acknowledgments:** The authors would like to thank the reviewers for valuable suggestions that increased the quality of this paper.

**Conflicts of Interest:** The authors declare no conflict of interest.

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
