# Peer review of "How Does the Geography Curriculum Contribute to Education for Sustainable Development? Lessons from China and the USA"

_sustainability, doi:10.3390/su141710637_

Round 1
Reviewer 1 Report
· The explanation which signifies the importance and the relevancy of Geography education with ESD or sustainability itself were not clearly stated. Author only provide a very brief and general information regarding Geography education which did not give any indication or highlight the role of Geography Education towards sustainable development compared to other subjects. By quoting past scholars regarding Geography education as ‘the science for sustainability’ is inadequate. Author should elaborate more on this concept, thus, to prepare readers the ‘identity’ of Geography Education which could promote this subject as the significant subject in promoting ESD.
· Please be consistent in using special terms. Mentioned here is ‘sustainable development education’, is this referred to ESD or is this another concept? Thus, consistency in using and referring to special terms must be observed.
· RQ1: What is the role of geographic topics in sustainable development? A topic of any lesson does not imply its role, but just merely stating the scope of the teaching and learning process of a lesson. Thus, intention of RQ1 is not clear.
· The overall description of China’s Geography Curriculum was not given except very brief Table 2 & Table 3, regarding Environmental related concepts – topics in Geography Curriculum in China. A background of the content should be included as supportive reference to the purpose of this paper which was stated as “This paper explores the role of the school geography curriculum in 89 China in developing learners’ awareness, attitudes and values of sustainable development 90 benchmarked against a recognized framework for sustainable development competencies 91 as measured against a number of indicators.”
· For the part on ‘Distribution of keywords’, the explanation was also confusing. Earlier in the paragraph (Line 584), author mentioned about profiling the documents and it was based on categories of terms; general education, science education processes, ecology, environmental education, environmental agency, sustainability terms and geography terms; but the result of the items as shown and explained have not been categorized according to these terms. Thus, what is the purpose of the above mentioned terms?
· Justification in choosing America’s Geography curriculum were not clearly stated and justified.
· ‘5.3. Learning objectives that reflect the content of ESD in America’ – no references at all? Thus, where did the author get the information from?
· The discussion did not mention the result between China and America’s geography curriculum content regarding ESD? Thus, what is the purpose of referring to America’s curriculum. Overall the discussion and the conclusion can be strengthened by giving more insights or author’s own perception regarding the curriculum and tied it up issues surrounding China and globally to signify the importance of the curriculum contents in supporting sustainable development. Repeating the result and making suggestion should be accompanied with the relevancy of the issues facing by the present and future generations.
Author Response
The explanation which signifies the importance and the relevancy of Geography education with ESD or sustainability itself was not clearly stated.
Response:
The explanation of the importance of Geography to ESD was stated on Line 162-172, and also in the Discussion part, from Line 481-487.
Please be consistent in using special terms. Mentioned here is ‘sustainable development education, is thus referred to as ESD, or is this another concept?
Response:
The special terms were consistent now, it means ESD.
RQ1: What is the role of geographic topics in sustainable development?
Response:
This question was revised in the part "5.1. The role of China’s Geography Curriculum for ESD" now.
The overall description of China’s Geography Curriculum was not given except in very brief Table 2 & Table 3, regarding Environmental related concepts – topics in Geography Curriculum in China. The background of the content should be included as a supportive reference to the purpose of this paper.
Response:
The overall description of China’s Geography Curriculum was now added, from Line 162-175, including the background of the content.
For the part on ‘Distribution of keywords’, the explanation was also confusing.
Response:
This part was revised now, from Line 263-279, with the result of the items categorized according to these terms.
Justifications for choosing America’s Geography curriculum were not clearly stated and justified.
Response:
This is clear now, in Line 122-124.
The discussion did not mention the result between China and America’s geography curriculum content regarding ESD? Thus, what is the purpose of referring to America’s curriculum?
Response:
The comparison of China and America’s geography curriculum content has been added in the discussion part now.
Thanks for your review report.

Reviewer 2 Report
The manuscript is worth publishing. However, there are a few things to be corrected:
1. Lines 608-611: Groups 2 and 3 were incorrect. There is no group for “development” and “sustainable”. Please check the dendrogram (Figure 2) to revise the two groups. One should cover nine keywords from “describe” to “illustrate”, while another one should cover “development” and “sustainable”.
2. Line 714: Please add Figure 4 in the first sentence. For example, you may revise the sentence as follows: In the American Geography Curriculum Standards (as shown in Figure 4), the topics that are most relevant to sustainable development include "humanity", "globalization", "community", "environment", "Earth", "resources" and "society".
3. Line 724: Please add Figure 5 in the first sentence.
4. Line 723: The concept of link analysis has not been described. Please add its description in the “4.1 Research Design” section.
5. Line 732: They should be Figure 6 and Figure 7.
6. Line 735: Why didn’t “illustrate”, “country”, and “characteristics” receive the most coding references, according to Figure 6?
7. Line 779: It should be Figure 7.
8. Line 780: Why aren’t “illustrate” and “country” the most frequent keywords, according to Figure 7?
9. Lines 1139-1140: Z.T. and Y.L. are not the authors of this manuscript, are they?
10. The manuscript has many careless and grammatical mistakes, especially newly revised parts (in red color). Professional copy-editing service may be required unless the authors will have checked all of these mistakes.
Author Response
- Lines 608-611: Groups 2 and 3 were incorrect.Response: They are correct now.
- Line 714: Please add Figure 4 in the first sentence. Response: Added Figure 4 in the first sentence.
- Line 724: Please add Figure 5 in the first sentence. Response: Added Figure 5 in the first sentence.
- Line 723: The concept of link analysis has not been described. Please add its description in the “4.1 Research Design” section. Response: Add the description of link analysis in the “4.1 Research Design” section.
- Line 732: They should be Figure 6 and Figure 7. Response: They are correct now.
- Line 735: Why didn’t “illustrate”, “country”, and “characteristics” receive the most coding references, according to Figure 6? Response: They are correct now, the frequency of “illustrate”, “country” and “characteristics” received most coding references.
- Line 779: It should be Figure 7. Response: It is Figure 7.
- Line 780: Why aren’t “illustrate” and “country” the most frequent keywords, according to Figure 7? Response: the most frequent keywords in China’s high school geography curriculum geography are “illustrate” and “explain”.
- Lines 1139-1140: Z.T. and Y.L. are not the authors of this manuscript, are they? Response: Yes, they are not the authors, they had participated in the coding progress.Thanks for your review report, changes have been made according to your suggestions.

Reviewer 3 Report
The authors could see my comments in the article!
The major problem is the structure of the article where the analysis of US American Geography Curriculum Standards did not follow the same steps as China's Geography Curriculum Standards analysis. There is no reference in American GCS neither in Discussion nor in Conclusion and there is no limitation of the study or the comparison of these two curricula.

Author Response
Thanks for your suggestions.
I have made more revisions according to your review report.
The structure of the article has been adjusted.
The analysis of US American Geography Curriculum Standards was added.
References in American GCS were also added to the Discussion part.
Add the limitation of the study or the comparison of these two curricula.

Round 2
Reviewer 3 Report
My suggestions are that the authors should add in the article a better analysis of American Geography curriculum standards. They could see my comments in the article.

Author Response
Dear reviewer,
I have made more revisions according to your comments, attached is the point-by-point response.
Thank you so much.

Round 3
Reviewer 3 Report
I agree on the article for publication
Author Response
Dear reviewer,
I have revised the manuscript, according to your helpful comments, using the "Track Changes" function so you can see the clearly highlighted revisions.
Thank you so much.

This manuscript is a resubmission of an earlier submission. The following is a list of the peer review reports and author responses from that submission.